# Model Class Reliance for Random Forests

**Gavin Smith**
N/LAB
University of Nottingham
Nottingham, UK

**Roberto Mansilla**
N/LAB
University of Nottingham
Nottingham, UK

**James Goulding**
N/LAB
University of Nottingham
Nottingham, UK

{first.last}@nottingham.ac.uk

## Abstract

Variable Importance (VI) has traditionally been cast as the process of estimating each variable's contribution to a predictive model's overall performance. Analysis of a single model instance, however, guarantees no insight into a variables relevance to underlying generative processes. Recent research has sought to address this concern via analysis of Rashomon sets - sets of alternative model instances that exhibit equivalent predictive performance to some reference model, but which take different functional forms. Measures such as Model Class Reliance (MCR) have been proposed, that are computed against Rashomon sets, in order to ascertain how much a variable must be relied on to make robust predictions, or whether alternatives exist. If MCR range is tight, we have no choice but to use a variable; if range is high then there exists competing, perhaps fairer models, that provide alternative explanations of the phenomena being examined. Applications are wide, from enabling construction of 'fairer' models in areas such as recidivism to health analytics and ethical marketing. Tractable estimation of MCR for non-linear models is currently restricted to Kernel Regression under squared loss [7]. In this paper we introduce a new technique that extends computation of Model Class Reliance (MCR) to Random Forest classifiers and regressors. The proposed approach addresses a number of open research questions, and in contrast to prior Kernel SVM MCR estimation, runs in linearithmic rather than polynomial time. Taking a fundamentally different approach to previous work, we provide a solution for this important model class, identifying situations where irrelevant covariates do not improve predictions.

## 1 Introduction

Post-hoc model explanation, through the use of variable importance measures, underpins a range of applications including variable selection, fit-for-purpose model auditing in domains such as criminal recidivism cases and the identification of potentially causal features. However, given input features typically share predictive information regarding the output variable, the learnt model potentially encodes just one of many equally well performing functional relationships between the input and output variables, with the choice often made arbitrarily with respect to the use case by the model building algorithm. This means variables may be incorrectly considered unimportant, that model audits are not robust to model retraining and the interpretation of potential causal features may be incomplete and misleading. In some cases these issues do not matter (i.e. minimal subset selection when predictive performance is the only goal) but often they do, with any machine learning framework/pipeline that subsequently includes the interpretation/explanation of a single model (e.g. [21, 6, 8, 5, 10]) likely to be improved.

Recently, [7] formally defined this issue - the multiplicity of models with the same prediction performance based on different underpining predicting factors - with regard to model classes. Given

a model class (model type and fixed complexity), the authors define a $\epsilon$-Rashomon set as the set of all equally performing (within a tolerance of $\epsilon$) models from a given reference model. Based on the definition of such sets the authors define two *Model Class Reliance* measures, MCR+ and MCR-. In contrast to global variable importance measures which aim to explain the degree to which a given model relies on (utilizes) each variable in the given model, MCR represents the most (MCR+) and least (MCR-) a variable is relied on in any model in the $\epsilon$-Rashomon set. The tractable provision of upper and lower bounds of variable reliance across an entire model class, as opposed to a single model, provides information able to be utilized within the model building phase and/or in a post-hoc fashion to better evaluation the optimally of the model and the existence of alternatives.

For instance in motivating MCR, [7] utilized MCR- and MCR+ to audit models/problem domains to examine the potential reliance of models on variables considered unacceptable, a problem particularly relevant in criminal recidivism prediction. A second application is with regard to all relevant variable selection [5]. While there are well known approaches based on heuristics for these problems [11], MCR+ has a notable use-case here in providing a simple and theoretically optimal solution since if the MCR+ is zero the variable cannot be relevant. A third application is as part of the model building pipeline, enabling the incorporation of domain knowledge to either realize better models or, if exploratory analysis is being conducted provide a better understanding of the phenomenon being modelled. For instance, model reliability may be improved by selecting models that highly utilize variables known or hypothesised to be causal over others. Alternatively practitioners may consider one model to be more actionable than another, for example where predictive models are used for targeted communications with the marketing domain. Here the model itself being used to select who to target and the explanation is used to inform communication strategy and/or design. Being aware of equally predictive models enables a more relevant choice based on knowledge of the specific problem not available to the model building algorithm.

While MCR and the underpinning concept of Rashomon sets offer significant potential, the initial work by [7] provides only a limited number of methods, for a restricted number of model classes. These include models that have convex loss functions in the model parameters and linear and Kernel SVM regression with squared loss functions. Notably the former is considered potentially intractable by the authors in the general case and the latter (for which they provide a RBF Kernel implementation with polynomial runtime complexity) is the only general purpose non-linear model class. Acknowledging the limitations directly, the authors leave as open problems more general purpose non-linear algorithms, particularly noting model classes in which irrelevant covariates do not improve predictions hypothesising these many be potentially intractable [7, pg. 19, Table 1].

In this work we directly address this gap, providing a method to compute MCR for both classification and regression Random Forests with linearithmic runtime complexity. The use of Random Forests additionally provides a higher utility in practice as they are comparatively robust to correlated and/or irrelevant variables, thereby often being a better choice of model class for the reference model. This is particularly true given (1) alternate explanations in part stem from correlated features and (2) the distinction between irrelevant vs. relevant variables is unknown when attempting to learn (alternate) explanations.

## 2 Comparison of MCR with existing VI methods

Model Class Reliance aims to measure fundamentally different (but related) information to existing variable importance techniques. Methods such as unconditional permutation importance [3] and recent variants [7] as well as methods such as Gain [2], Split Counts [4] and Minimal Depth [9] which leverage the structure of tree based models seek only to measure the importance of a variable with regard to a single predictive model. Recently Minimal Depth was extended by [18] such that a variable's importance, measured by its depth, also included when it could have been used to realize a split identically to the chosen split. This can be seen as enabling a single tree to represent many others from the same class (and our approach leverages this insight in part). As will be discussed, however, this is not enough to achieve tight MCR bounds. Moreover, the approach by [18], being a measure based on tree structures is known to be inconsistent [15] reporting incomparable attribution scores as models change their structure making the results difficult to interpret in many use cases.

SHAP values [14, 15] by design measure a variable's average contribution and therefore are not suitable for measuring contribution bounds. Specifically, seeking the maximum or minimum contribution

for a variable breaks the first property, local accuracy, from which SHAP is derived. We note that this is not deficit of SHAP, SHAP fulfils its aim to explain a specific model instance, but a fact that highlights that SHAP is unable to be re-purposed as an MCR measure. Algorithm Reliance (AR) [7] methods retrain models to determine the effect of holding out variables. [7] consider two types of AR, *AR necessity* and *AR sufficiency*. The former considers hold out of the variable of interest, and the latter the holding out of all variables except the one of interest. While *AR necessity* has conceptual similarities to MCR-, AR methods provide a different type of measurement, considering a larger class of models, and including candidates that do not fit the data equally well [7][1]. Moreover, the computational requirement to retrain a model per variable of interest can render them impractical.

Conditional variable importance [20] is a related permutation importance measure which disrupts a variable conditional on full information of all other variables. If information is shared between the variable of interest and other variables this reduces the impact of the permutation to the appropriate degree. Therefore, as long as the conditional distribution can be correctly approximated conditional permutation importance provides a similar measure to MCR-. The fact that it is based on a single model is not important here as the fact that a variable could have been used in a different model in the class is irrelevant, and it is desirable that no effect is recorded for this variable. In practice, however, approximating the conditional distribution can be error prone in the case of real valued datasets.

The most related work to MCR+ is [1, 16] which seek to identify *indirect influence* for black-model auditing. In contrast to MCR+ these methods are designed to focus solely on auditing a single model by, for a given variable of interest, 'noising up' the other input variables to remove any shared information and then measuring the loss in predictive performance. The proposed algorithms learn to disrupt relationships between the variable of interest and input variables independently and current approaches are either overly simplistic (considering only pairwise variable relationships [1]) or required significant computation time and the tuning of complex of meta-parameters [16] resulting in different levels of error in the learnt relationship per variable. Without significant effort in tuning the algorithms, the potential for differing levels of error confounding the reported variable's importance make their interpretation somewhat tenuous. In summary, while related, the methods are both not designed to address the same problem as MCR+ and are currently complex to use in practice.

## 3 Model Class Reliance Preliminaries

In this section we introduce the notation required for us to define MCR estimators for Random Forest model classes. In general, we will broadly accord to that used in [7], in referring to a predictive model, $F \in \mathcal{F}$, that has been constructed using a dataset generated from a random variable $Z = (X_1, X_2, Y) \in \mathcal{Z}$, where our target variable is $Y \in \mathcal{Y}$, and our predictors are sets of covariates $X_1$ and $X_2$. These input variables are separated merely to allow us to focus on $X_1$ in notation as the (potentially multivariate) variable whose importance is being investigated, with $X_2$ being the remaining input features. To derive our estimators, we first define several key terms:

**Model Reliance:** reflects the extent to which a single fitted model relies on variable $X_1$ to achieve its predictive performance, and is denoted as $MR_{X_1}(F)$. To calculate model reliance, we first apply a user-defined non-negative loss function $L : \mathcal{F} * \mathcal{Z} \to \mathbb{R}_{\geq 0}$ to assess the model's predictive performance. Next we disrupt $X_1$ using some specified function, $\Phi : \mathcal{Z} \to \mathcal{Z}$, in order to render the variable as uninformative as possible for prediction, while keeping the same marginal distribution. The new dataset after disruption, $Z^{\Phi} = (X_1^{\Phi}, X_2, Y)$, is then used to obtain a new expected loss score, $\mathbb{E}L(F, Z^{\Phi})$. Model reliance is then derived as either the ratio or the difference between the two expected loss scores (before/after disruption) [7]. In this study we have used the difference:

$$MR_{X_1}(f, \Phi) = \mathbb{E}L(f, \langle X_1^{\phi}, X_2, Y \rangle) - \mathbb{E}L(f, \langle X_1, X_2, Y \rangle) \tag{1}$$

Model reliance can be interpreted as the loss introduced when we destroy $X_1$'s predictive capacity. However, final interpretation relies on the choice of disruption function, $\Phi$. MR was introduced in [7] using a 'switch' permutation strategy, $\Phi_S$ where all possible permutations of $X_1$ with $\langle X_2, Y \rangle$ that do not exist in the original dataset, $Z$, are exhaustively constructed to form $Z^{\Phi}$. However, the notion of model reliance is generalizable to other disruption strategies (e.g. unconditional permutation [3]). Note also that the size of the disrupted dataset, $Z^{\Phi}$, will be greater than or equal to the original given

that $|Z^{\Phi}| = n|Z|$, where $n$ reflects the number of permutation rounds specified when $\Phi_U$ or $\Phi_C$ are used, or exactly $n = |Z| - 1$ for the exhaustive permutation strategy used by $\Phi_S$.

**Rashomon Set:** Model reliance tells us only about the contribution $X_1$ makes to a single model instance. To understand model reliance across a whole class of predictors, we must examine the range of MR values which exist across all equally well performing predictors. This is the $\epsilon$-Rashomon set, and given a reference model $F$, is the set, $\mathcal{R}_\epsilon(F)$, of all models with a predictive performance within $\epsilon$ of $F$ [7]. For most practical applications $F$ refers to the best performing model found via cross-validation (or some equivalent testing procedure).

**Maximum Model Class Reliance ($MCR^+$):** The maximum possible change in predictive performance that could be attributed to our variable of interest, $X_1$, in any model within the $\epsilon$-Rashomon set, $R_\epsilon(F)$. Note that $MCR^+$ corresponds to the MR of a realizable model, and is defined as:

$$MCR^+_{X_1}(F, \Phi) = \max_{F' \in R_\epsilon(F)} \left( MR_{X_1}(F', \Phi) \right) \tag{2}$$

**Minimum Model Class Reliance ($MCR^-$):** In a similar manner, we can examine the minimum possible change in predictive performance attributable to $X_1$ for any model in the $\epsilon$-Rashomon set:

$$MCR^-_{X_1}(F, \Phi) = \min_{F' \in R_\epsilon(F)} \left( MR_{X_1}(F', \Phi) \right) \tag{3}$$

**Empirical MCR ($\widehat{MCR}$):** The range, $MCR$, formed from $[MCR^-, MCR^+]$ of course refers to population level statistics, which we cannot obtain with our sample dataset, $Z$. Thus estimators for all of the above must be used in practice, with MR being estimated using our in-sample loss rather than expected loss over the whole population. This is referred to as our empirical MCR:

$$\widehat{MCR}_{X_1}(F, \Phi) = \left[ \min_{F' \in \hat{R}_\epsilon(F)} \left( \widehat{MR}_{X_1}(F', \Phi) \right), \max_{F' \in \hat{R}_\epsilon(F)} \left( \widehat{MR}_{X_1}(F', \Phi) \right) \right] \tag{4}$$

## 4 A novel $MCR$ method for Random Forests

Although an $\widehat{MCR}$ range is fully determined by a sample, actually computing it in practice is highly non-trivial, mostly due to the maximization and minimization steps required in equations 2 and 3 respectively. This challenge was recognized in [7], which introduced a polynomial-time optimization procedure to estimate MCR for Kernel-based SVM classes. We now present a novel method for estimating $MCR$ for Random Forest model classes, that functions in linearithmic time.

Recall that we seek to find the maximum class reliance, $MCR^+_{X_1}(F)$ and minimum, $MCR^-_{X_1}(F)$, across all models in an $\epsilon$-Rashomon set, $\mathcal{R}_\epsilon(.)$, with reference to some variable of interest, $X_1$. The Rashomon set is itself established based on a preexisting reference model instance, $F$, of some fixed complexity, $\theta$, and trained using data, $Z = \langle X_1, X_2, Y \rangle$. Here $X_2$ represents additional input features accompanying $X_1$, and $Y$ the output feature. We reasonably assume it is impossible to iterate over every element in $\mathcal{R}_\epsilon(F)$ to obtain true MCR values directly due to computational intractability.

We therefore instead seek estimates for $\widehat{MCR}^+(F)$ and $\widehat{MCR}^-(F)$ that are both tight and obtainable within reasonable time complexity. In order to derive such estimators, we take a divide-and-conquer approach, conceptually mapping each tree in the reference model to a new tree which maximises (or minimizes) reliance on $X_1$ while maintaining predictive equivalence with to F (and hence membership of the Rashomon set, $R_\epsilon(F)$). Our strategy for achieving this traverses an estimate of each individual tree's own Rashomon set, $\hat{R}_\epsilon(f)$, via application of two key transformations, as follows:

### 4.1 Estimation of $MCR^+$

For each tree in the forest, $f \in F$, its individual Rashomon set, $R_\epsilon(f)$, is examined to find a new tree that is maximally reliant on $X_1$ yet has identical structure and makes the same predictions across all data (a characteristic henceforth referred to as *structural equivalence*). Despite the strictness of this mapping, searching the whole of $R_\epsilon(f)$ is computationally intractable, so we instead apply a transformation function, $\mathcal{S}^+(f)$ that leverages *surrogate splits* [2]. This function examines each node in $f$, and replaces the existing decision variable with $X_1$ if they are surrogates (identified

during construction of $F$ [18]). This allows us to directly identify a preferred tree in $R_\epsilon(f)$, from the $m^{\frac{n}{2} \leq n\_splits \leq n-1}$ structurally equivalent trees to $f$ that exist [13] where $m$ is the number of input features and $n = |Z|$ is the number of data points. We are guaranteed due to this structural equivalence that:

$$\forall f \in F : \widehat{MR}_{X_1}(\mathcal{S}^+(f)) \geq \widehat{MR}_{X_1}(f) \tag{5}$$

Once this transformation has been applied to all trees, we can construct an entirely new random forest, $F_S^+ = \{S^+(f) : f \in F\}$. By construction $F^+$ will produce exactly the same predictive outputs as $F$, despite containing components with greater usage of $X_1$, and hence:

$$MR_{X_1}(F_S^+) \geq MR_{X_1}(F) \tag{6}$$

Having formulated $F_S^+$, the next step in estimating $MCR^+(F)$ is to relax the requirement for structural equivalence altogether. To implement this we identify a random forest in $R_\epsilon(F)$, that uses trees from $F_S^+$ but in a new combination (and allows for duplication) that maximizes reliance on $X_1$ but which maintains the same prediction accuracy.

This is achieved via a second transformation $T^+(F_S^+)$, mapping each tree in $F_S^+$ to some other in $F_S^+$ that produces the same predictions but which leverages our variable of interest the most (n.b. if a tree is already the most reliant on $X_1$ it is mapped to itself). To formalize this, let us first define $\Psi(f)$ as the set of trees that share *predictive equivalence* with $f$, given dataset $Z$:

$$\Psi(f) = \{f' \in F : \forall_{x \in Z} \ f(x) = f'(x)\} \tag{7}$$

Note that this notion of predictive equivalence between models, $\Psi$, doesn't require they share any structural similarities whatsoever, nor that their predictions even be correct - but merely that they are the same for all data in $Z$. Given this, we can succinctly define $T^+(f)$ as:

$$T^+(f) = \underset{f' \in \Psi(f)}{\arg\max} \ MR_{X_1}(f') \tag{8}$$

Applying this transformation to map every tree to another in its set of predictive equivalents that has maximal $MR$, generates a co-domain that we can collate to construct our boundary model:

$$F_T^+ = \{T^+(f) : f \in F_S^+\} \tag{9}$$

This model remains in the Rashomon set of $F$, and importantly we remain guaranteed that $MR_{X_1}(F_T^+) \geq MR_{X_1}(F_S^+)$ (please see supplementary materials for the full proof). To obtain our our final estimator for $MCR^+$ we simply assess this model's reliance on $X_1$:

$$\widehat{MCR}_{X_1}^+(F) = MR_{X_1}(F_T^+) \tag{10}$$

With $\widehat{MCR}^+$ in hand, the question remains as to why this reflects a good estimator of the class' maximum reliance on $X_1$. To demonstrate this, consider first that as the number of trees in $F$ tends to infinity, all possible constructable trees will occur in its ensemble (due to the stochastic nature of its generating algorithm). This means the transformation $T^+(f)$ will be replacing every tree that can possibly exist, with the tree from its complete Rashomon set that holds maximum reliance on $X_1$ for any potential composition of forest. Thus, as $|F| \to \infty$, our estimator $\widehat{MCR}^+(F)$ can only increase monotonically towards $MCR^+$. However, given we only ever have finite models in practice the question remains - *how quickly does it increase?* The answer to this is to some extent dependent on the size of the dataset, |Z|, and the reference tree's complexity constraints, $\Theta$. However, we will show empirically in §5.3 that for practical purposes our method converges to a consistent asymptotic value for the estimator for only a moderate numbers of trees. This will provide evidence that a good estimator is likely to been obtained in the majority of cases. On synthetic data where the solution is known we will further show that the model is able to recover the true $MCR^+$ value almost precisely.

## 4.2 Estimation of $MCR^-$

An estimator of $MCR^-$, the minimum model class reliance on $X_1$, can be derived in similar fashion to that demonstrated in §4.1. However, in this case we first apply transformation $\mathcal{S}^-(f)$ to every

tree in the reference model, $F$, a mapping that again implements surrogate splits, but which removes (rather than injecting) $X_1$ from any decision node when a surrogate for it is available. This guarantees:

$$\forall f \in F : \widehat{MR}_{X_1}(\mathcal{S}^-(f)) \leq \widehat{MR}_{X_1}(f) \tag{11}$$

Once applied to all $f \in F$, we can again conceptualize a forest that contains only these minimally reliant trees, $F_S^-$. We then apply a second transformation, $T^-$ (again in symmetry with our derivation of $MCR^+$), that substitutes any tree in $F$ with another that is in its set of predictive equivalent trees, $\Phi$, but which has the least reliance on $X_1$:

$$T^-(f) = \underset{f' \in \Psi(f)}{\arg\min}\ MR_{X_1}(f') \tag{12}$$

Once this transformation is applied to all trees in $F_S^-$, we are left with a model, $F_T^-$, which remains firmly in the reference model's Rashomon Set while having minimal reliance on $X_1$. As before our our final estimator $MCR^-$ can then be calculated as this boundary model's overall reliance on $X_1$:

$$\widehat{MCR}_{X_1}^-(F) = MR_{X_1}(F_T^-) \tag{13}$$

## 4.3 Computational of RF-MCR in practice

The computational complexity for computing RF-MCR is $\Theta(K\tilde{N}log_2(N))$ for both regression and classification. $\tilde{N}$ is the size of the set MCR is being evaluated on (training or a held out set). This assumes the reference model is built using a modified Random Forest algorithm in which surrogates are recorded (requiring all variables to be considered as potential splits at each node even if they are not being considered as a main split) and the final instance predictions per tree used to record prediction equivalent tree groupings ($\Psi$, Eqn 7), costing $kn$ operations using a hashmap. The latter requires bootstrapping be avoided and randomness introduced into the forest via variable split selection. Complexity is unchanged from existing algorithms with these constraints: $O(mkn\ log_2 n)$.

Computation of RF-MCR is done in 3 steps. Step 1 computes the $MR(S^+(f))$ for each tree by computing unconditional VI but with forced surrogate use (MCR+ or forced avoidance, MCR-) with regard to the variable of interest. This involves making a prediction per data point and can be done in $K\tilde{N}log_2(N)$. The result is the set $F_S^+$ and associated $MR$ scores. Step (2) leverages $\Psi$ and $MR(S^+(f))$ to compute $F_T^+$ (Equation 9). This can be done in $K$ time by iterating over the groups of trees that are predictive equivalent and, as a replacement for all trees in that group, selecting the one with the highest $MR$. The final step (3) involves computing $MR(F_T^+)$ which takes $K\tilde{N}log_2(N)$.

So far only the computation of 0-Rashomon sets has been considered. The approach can be extended to a $\epsilon$-Rashomon set by: (1) relaxing the requirement for surrogate splits to be exact [18] and (2) the relaxation of exact prediction equivalence when forming $\Psi$. For (1) is not clear how this could be easily linked to a desired $\epsilon$ value and as such this extension is left as future work and only extension (2) implemented in this work. This does not introduce any additional runtime complexity.

Finally, as noted in Section 3, the proof for Empirical MCR is based upon in-sample loss, rather than the expected loss over the whole population. This leaves an outstanding question as to whether population level MCR+/- values can be accurately determined by constructing/searching over $\epsilon$-Rashomon sets identified from a sample. To this end [7] provide an applicable theoretical proof, demonstrating that in-sample estimation of MCR is sufficient under large sample sizes, assuming a reference model with a correctly selected complexity [7, §4.1]. This proof corresponds to the real-world use case where MCR is to be computed based on a reference model that has been trained on the full dataset (following the frequent strategy of applying cross-validation in order to determine a model complexity, prior to refitting on all data to produce final model outputs). In particular, this approach is commonplace when computing traditional variable importance measures (i.e. via permutation) [17]. A thorough investigation into the use of MCR within a sample splitting approach [7] remains interesting future work.

# 5 Empirical Evaluation

In this section we demonstrate: the ability of RF-MCR[2] to compute $\widehat{MCR}$ bounds in a tractable fashion; the convergence of the estimators; the ability to recover true $MCR$ values; and the insights derivable on 2 real world datasets (recidivism and breast cancer). A third real-world application (negative birth outcomes in East Africa) is additionally considered in the supplementary material.

First we consider data sampled from a simple generative network, to show ability to recover true MCR bounds. The network is setup such that variables $A$ and $B$ form an XOR predictive relationship, meaning *both* are required to predict $Y$ above random chance (50% accuracy). If both are known, then a perfect prediction can be made. Additionally, a variable $C$ is generated so as to be perfectly colinear with $B$. The true MCR range (with MR measured as mean decrease in accuracy) for $A$ is [50%, 50%], and $B$ and $C$ both have true MCR of [0%,50%]. A dataset of 1000 instances is generated from this network, to which various variable importance approaches are applied alongside RF-MCR. As this data forms a non-linear *classification* problem, SVM-MCR [7] cannot be applied.

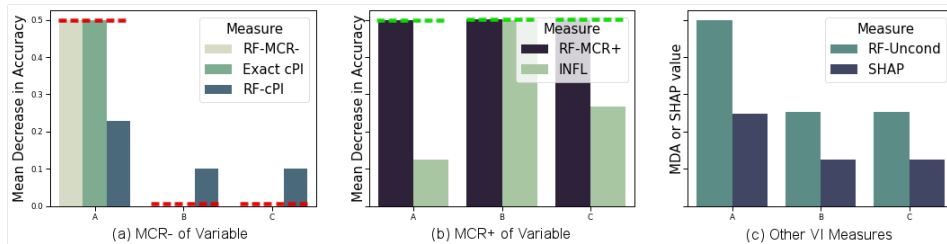

Figure 1: Synthetic Dataset Results: (a) $MCR^-$ recovery (b) $MCR^+$ recovery (C) VI methods. True values or MCR- and MCR+ are indicated in red and green respectively

As shown in Figure 1(a), both our approach and conditional permutation importance (Exact-cPI) generate values that correspond perfectly to the true $MCR^-$. The approximation of cPI for conditional random forests (RF-cPI) [20] is also shown, although its approximation errors are evident. Figure 1(b) assesses the identification of MCR+, comparing RF-MCR to the previously mentioned disentanglement technique in [16] (which we refer to as INFL here). Again, RF-MCR recovers the true values for each variable correctly. However, the neural networks used by INFL seem unable to correctly learn the underlying relationship despite significant tuning (full details of tuning are in the Supplementary Material), significantly underestimating the ability of A and C to contribute to successful predictions. Finally, Figure 1(c) shows that two other VI approaches, SHAP and unconditional permutation importance, offer no insight into $MCR^-$ and $MCR^+$ as expected.

These results provide initial evidence that RF-MCR can recover true model class reliance values - and the inability of traditional VI methods to reflect this range. However, they do indicate some correspondence between cPI and INFL with $MCR-$ and $MCR+$ respectively. In the following experiments we combine these two measures to form a HYBRID-MCR method solely for purposes of comparison. For regression, where MCR for Kernel Regression under squared loss (SVM-MCR) [7] has been previously proposed, we compare to both HYBRID-MCR and SVM-MCR.

## 5.1 COMPAS - Recidivism Modelling

Next we examine the effectiveness of RF-MCR on a real world regression dataset that aims to predict the sensitive issue of recidivism (probability of re-offending) using the same dataset as [7]. We replicate their results for SVM-MCR on the COMPAS data, and extend their analysis from SVMs to Random Forest classes. The COMPAS system provides records from a set of defendants from Broward County, Florida labelled with known recidivism scores [12]. The goal is to understand if unfair and potentially inadmissible variables, such as race, might be being unfairly relied on by models. Assuming a reference model of sufficient predictive performance can be obtained, identifying a high $MCR+$ in analysis tells us a variable may be being heavily leveraged in predictions; a low $MCR^-$ score squarely indicates proxies for $X_1$ exist, and its use need not be the case.

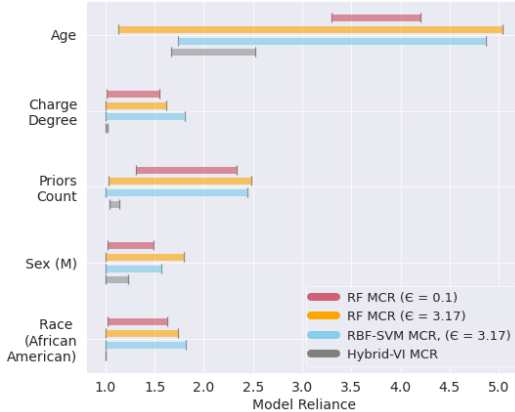 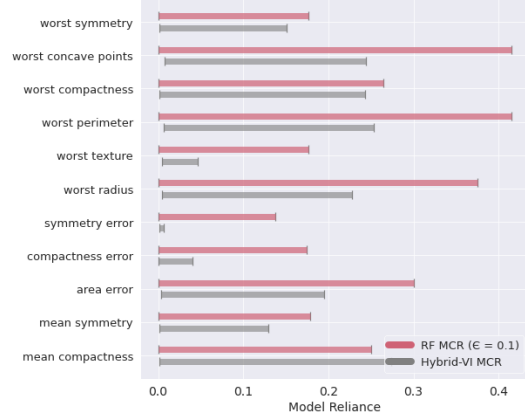

Figure 2: COMPAS dataset MCR Results          Figure 3: Breast Cancer dataset MCR results

Figure 2 shows the results. Both SVM-MCR and RF-MCR methods utilize reference models with similar mean squared errors (2.86 and 2.73 respectively) on a held out test set. In practice the SVM-MCR requires $\epsilon > 0$, so is set at $\epsilon = 3.17$ as per [7]. RF-MCR is tested at $\epsilon = 0$ (where it can be compared to the unparameterizable Hybrid-MCR), but we additionally test RF-MCR at $\epsilon = 3.17$ for direct comparison with SVM-MCR. Despite using a definition of MR in §4 as the difference between model error before and after variable disruption (which we find more intuitive), we report results using their ratio to aid direct comparison with the results in [7]. Following [7] we also left truncate results, and to aid comparison of approaches present results all input variables individually.

The ranges in Figure 2 immediately highlight the similar results for SVM-MCR and RF-MCR (at $\epsilon = 3.16$). There are differences, as SVM and RF classes afford different model solutions, but their general agreement adds support to the fact that both techniques are providing good estimators of true MCR (for more see §5.3). Most importantly, we now have 2 model classes supporting the fact that race is a non-necessary variable in recidivism modelling. However, RF-MCR adds increased insight that age can also shares mutual information with other variables that might be used instead. The difference between age MCR for $\epsilon = 3.17$ and $\epsilon = 0$ is stark, but illustrates that as the Rashomon set shrinks, so does MCR range - with some variables being far more sensitive to this effect. Our Hybrid-MCR baseline illustrates the limits of using VI measures. The fact its lower bound is similar to SVM-MCR is interesting (and requires further analysis), but it clear that the upper bound, INFL [16] is a poor estimate - not unexpected given it was not built for this purpose, but emphasizing the value of MCR to obtain domain insights into an underlying domain.

## 5.2 Breast Cancer Dataset

The Breast Cancer Wisconsin dataset [19] is a dataset from the UCI ML Repository, well known as having input variables with shared information. The dataset reflects a classification task to predict a tumour as malignant or benign. While well examined, this is the first time model class reliance could be applied to the dataset. Results in Figure 3 highlight the wide MCR ranges that exist across almost all variables, emphasizing the ability of RF-MCR to rapidly identify non-linear correlations. No variable is uniquely predictive within it; and only a few variables (with small ranges) offer little contribution to the predictive task at all. Given the good predictive performance of resulting models (95.74%), this means that multiple, distinct combinations of variables can produce equally performant models. For each feature, there exists some model that achieves maximal predictive performance where that feature is not required. While MCR currently provides overarching insights of this nature, determining the minimal subsets of variables for which equally performant models exists remains interesting future work. MCR analysis of this data does, however, demonstrate that when variable importance analysis is applied to a single model misleading results can occur - especially if one is looking for insight into some underlying generative process. In such cases, one of many predictive relationships can be overemphasized - hiding a variable of potential importance form practitioners, and/or conveying to them an incorrect understanding of the problem domain itself.

Comparing the RF-MCR to other baselines we note that no other non-linear MCR method exists for classification, so we can only contrast to our Hybrid-MCR baseline, which severely underestimates in its attempt to bound MCR+ (but please see supplementary material for error graphs, which indicate a relatively good disentanglement model was obtained, despite significant per feature variation in the noise and non-trivial reconstruction errors).

## 5.3 RF-MCR Analysis

Demonstrating the applicability of RF MCR we provide an empirical evaluation of runtime on the COMPAS dataset comparing to the polynomial time RBF-SVM MCR. The results are summarized in Figure 4(a) (see Supplementary material for more detail of results and setup) and report the time to compute the MCR range for an individual variable in seconds as the size of the dataset (train/test of equal size) is varied. The results clearly show the linearithmic nature of the implementation.

In Section 4.1 it was theoretically motivated that RF-MCR will quickly increase towards MCR+ as the number of trees tends to infinity (and vice-versa for MCR-). The question remains as to how quickly this occurs in practice. Figure 4 (b) and (c) provide empirical evidence for this for MCR- (COMPAS dataset) and MCR+ (Cancer dataset). The full set of MCR-/+ graphs are included in the Supplementary Material and reflect similar results - that as expected quick convergence to a point of stability for all variables is obtained with a moderate number of trees.

Finally an investigation was performed into the effect of the two transforms: the use of surrogate splits and prediction equivalence of trees by comparing the MCR results to an ablated version where prediction equivalence trees were not considered. Traditional permutation importance on the reference model used as the baseline. The results showed that the use of surrogates identified between $1.11 - 15.38\%$ of the increase in the reported upper bound importance (MCR+) with prediction tree equivalence responsible for the rest. For MCR- the decrease attributed to the use of surrogates was between $2.06 - 100\%$ with prediction tree equivalence responsible for the rest. Full details and results are included in the supplementary material. The results empirically demonstrate the requirement for both components of the algorithm.

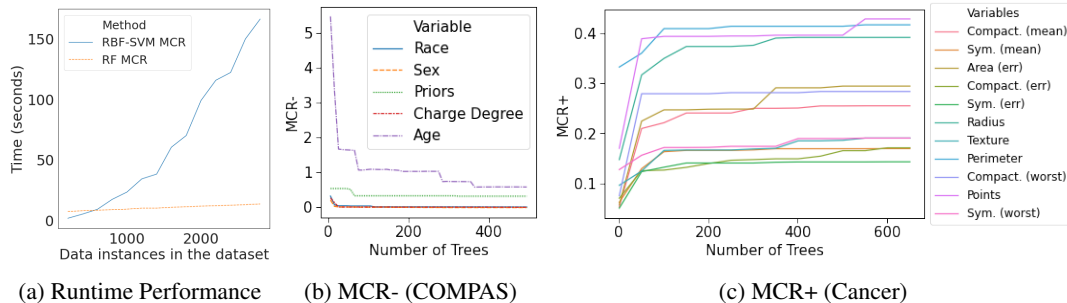

(a) Runtime Performance      (b) MCR- (COMPAS)      (c) MCR+ (Cancer)

Figure 4: RF-MCR Analysis Results

## 6 Conclusion

Model Class Reliance (MCR) provides a more comprehensive measure of variable importance than traditional methods. Considering all models within a model class, rather that the one model the model building algorithm realised, reveals important information about possible predictive relationships between a model's input and output features that practitioners may want to leverage or avoid.

In this work we significantly extend the utility and applicability of MCR in practice, introducing novel algorithms to estimate MCR for the Random Forest classification and regression model classes for the first time. In contrast to the only other known method for computing MCR in the non-linear case (Kernel SVM Regression), the proposed approach (1) includes non-linear classification for the first time and (2) significantly reduces run-time complexity from polynomial to logarithmic time enabling its use on large datasets. Evaluations on one synthetic and two real world datasets demonstrates the utility of the approach in practice.

## Broader Impact

The proposed work has the potential for significant societal impact. Understanding alternative predictive relationships that machine learnt models could equally be employing in their decision making while still achieving the same predictive performance is important for many applications. One particular case is when machine learnt models are then used in a way that significantly affects people's lives, such as the recidivism example detailed in this work. As such the approach has a part to play in ensuring algorithm fairness and reducing bias, including in model and data auditing. Other important use cases include scientific exploration of potentially causal factors in large datasets which then can form hypotheses for future causal studies. Without such methods proposed here that can be executed tractably and easily by those outside the field of computer science, the use of machine learning models may lead to misunderstanding if a practitioner assumes the relationship identified by a model is the only one or concludes that their hypothesised relationship does not exist or is not well performing when it is. A similar argument can be made with regard for the need for such techniques in business applications. Again as is discussed in the paper, the approach has applications for marketing and communication. The work could of course be used negatively. Rather than identifying and selecting the model from the set of best models in a principled way that benefits humanity, people could use it to select a model that furthers their aims at the detriment of others while still being able to claim they selected a performance optimal model. When releasing the model in a proprietary package this may go unnoticed.

## Acknowledgments and Disclosure of Funding

This work was supported by the following grants: Risk prediction for Women's Health and Rights in Tanzania: novel statistical methodology to target effective interventions (EP/T003928/1) and From Human Data to Personal Experience (EP/M02315X/1).

## Footnotes

[1]The difference between AR and MCR is discussed in more detail in the Supplementary Material.

[2]A python version of the implementation is available on github: `https://github.com/gavin-s-smith/mcrforest`.

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
