[Supplementary Material]

# Supplementary Material: Model Class Reliance for Random Forests

**Gavin Smith**
N/LAB
University of Nottingham
Nottingham, UK

**Roberto Mansilla**
N/LAB
University of Nottingham
Nottingham, UK

**James Goulding**
N/LAB
University of Nottingham
Nottingham, UK

`{first.last}@nottingham.ac.uk`

## 1 Implementation Availability

Our RF-MCR method for both Classification and Regression Random Forests is available as a pip installable python3 package on github: `https://github.com/gavin-s-smith/mcrforest`

## 2 Empirical Evaluation

All experiments were run from a Google Colaboratory notebook connected to a local instance utilizing an Intel® Core™ i7-3930K CPU @ 3.20GHz with 64GB RAM and running Ubuntu 20.04, Python 3.6.10 and R 3.6.3. Unless otherwise specified all algorithms were timed on single core versions even though, for instance, the proposed method is in places trivially parallelizable (i.e. during forest build). An exception was the grid search across meta-parameters to find the best (optimal) reference model where parallelization was used when required as this stage does not form part of the time comparisons.

### 2.1 Notes for Replication of Results

Replication is facilitated through the provision of four hosted Python notebooks which replicate the paper results. Hosted on Google Colaboratory they enable the use of hosted or local runtime environments. When tested hosted runtimes were running Python 3.6.9 and R 3.6.3. Please note that while a hosted runtime can be used for ease of replication, all timings reported in the paper were based on using a local runtime environment as previously indicated NOT a hosted environment.

The URLs for the notebooks are:

1. Synthetic Experiments:
   `https://colab.research.google.com/drive/1nHaP8iNnTyY8txptNASWT2fjRPvrOSy4`
2. COMPAS Experiments:
   `https://colab.research.google.com/drive/1JiJpC8KJAeALt1wxmluUWCCGnkWfsZCG`
3. Breast Cancer Experiments:
   `https://colab.research.google.com/drive/1VYzSf9vg6-kzQHJwdi-vUrwvjr-djqjQ`
4. RF-MCR Analysis:
   `https://colab.research.google.com/drive/1BFzoR9SDNRcKv1R9BX7dnRMcIG83TYdX`

The notebooks, when run in the hosted environment will automatically install the required packages developed as part of this work. The packages developed as part of this work are discussed below and made available via the above notebooks. Note that the learning of meta-parameters for the Disentangling Influence method from [4] is generally to time consuming for a hosted notebook and the parameters from a prior run have been included. By default the notebooks will use these and skip the meta-parameter search.

### 2.1.1 The proposed method RF-MCR source code

The code is written as an extension to the sklearn RandomForestRegressor and RandomForestClassifer classes. To simplify installation, the implementation has been decoupled from sklearn and packaged as a separate installable library. The code is available on github: `https://github.com/gavin-s-smith/mcrforest`

Dependencies, download links and install instructions are included in the notebooks in the case a local install is desired. If running the notebooks on a hosted instance this will be automatically installed.

### 2.1.2 A Python wrapper for the non-linear RBF-SVM MCR method from [1]

The wrapper calls the R code from the lead author's github `https://github.com/aaronjfisher/mcr` and derived from `https://github.com/aaronjfisher/mcr-supplement`.

Dependencies, download links and install instructions are included in the notebooks in the case a local install is desired. If running the notebooks on a hosted instance this will be automatically installed.

### 2.1.3 A Python wrapper for the Disentangling Influence method from [4]

The wrapper is based on the authors code from `https://github.com/charliemarx/disentangling-influence`. The wrapper extends the code base from the author to:

- include code to grid search for the neural network meta-parameters to find the optimal parameters per (encoder, discriminator, decoder) triple, noting that there is one triple per feature of interest. This code is parallelized. For the encoder and decoder neural networks two layers were used (reflecting [4]). The grid search considered 80 different model meta-parameterizations:
  - four values for the latent dimension equally spaced between the number of input features and the cardinally of output.
  - four values for each of the hidden layers in the encoder and decoder selected to be equally spaced between the latent dimension and two times the number of covariates. For the encoder, combinations of these for the two layers such that the first layer had more neurons than the second were considered. For the decoder reversed combinations were considered.
  - two different learning rates were considered 0.001 and 0.01
  - beta was set at 0.5 or 1 based on discussion in [4] and preliminary experimentation
  - the maximum number of training steps was set to 10000
- include early stopping to reduce runtime. Tolerance was set to 0.00001 and patience to 100.
- provide a simple object to simply the following:
  - (re)fit a set of triples based on a set of meta-parameters for all variables of interest. This simply calls the code from `https://github.com/charliemarx/disentangling-influence`.
  - computation of the influence value. This calls the code from `https://github.com/charliemarx/disentangling-influence` however, following [4] the off-the-shelf use of SHAP as the underpinning direct influence function is replaced by unconditional permutation importance. In addition the reporting of influence as either a difference or a ratio is coded.
  - a method to replicate the diagnostic graphs shown in [4] from the output is included

Dependencies, download links and install instructions are included in the notebooks in the case a local install is desired. If running the notebooks on a hosted instance this will be automatically installed.

### 2.1.4 Source code for a modified version of the R package party

A modified version of `https://cran.r-project.org/web/packages/party/index.html` as an installable package has been made. The modification is minor, slightly altering the varimp function (in varimp.R) to, on request, return the conditional permutation importance as a ratio instead of a difference. The party version modified was 1.3-4.

Dependencies, download links and install instructions are included in the notebooks in the case a local install is desired. If running the notebooks on a hosted instance this will be automatically installed.

## 2.2 Synthetic Experiments

The following implementation points are of note beyond those detailed in the paper:

- The reference model for the RF MCR is a model trained as required by the MCR method (no bootstrapping, a small number of variables considered at each split, in this case 1 given there are only three variables). The model is able to learn the relationship perfectly.

- Conditional Variable importance (Random Forest version) used the implementation from the R package party `https://rdrr.io/cran/party/man/varimp.html`. Default parameters were used.

- The disentanglement (INFL) method had it's meta-parameters optimised via the previously discussed grid search method. This learns an optimised model per input variable. Despite this significant tuning (and some further manual tuning attempts), the method was relatively unstable leading to a number of solutions depending on the different random seed used. Figure 1 shows the the diagnostic graphs as considered in [4]. They indicate that the resulting networks are relatively acceptable for the run reported in the paper. With respect to interpreting these graphs, [4, pg 6] note that: *Optimal is reconstruction error and prediction error of 0 for all features (indicating no errors in autoencoding), and disentanglement error of 1 for all features*. We note that some of the alternative solutions for different seeds observed corresponded to a slightly better performance of the method in the context of the paper, though with worse diagnostic graphs. In all cases the discussion regarding this method in the paper holds regardless of the exact observed result during experimentation. Note that the notebook does not have a fixed seed and this instability can be explored by re-running the notebook.

- SHAP values are calculated on an identical RandomForestClassifier as used for the RF MCR.

- Unconditional and exact conditional permutation importance is computed on the same RandomForestClassifier as RF MCR.

Figure 1: Graph of diagnostic measures for the neural networks trained as part of the Disentangling Influence (INFL) method from [4] for the Synthetic experiments. Graphs indicate the networks are reasonably well trained according to [4].

## 2.3 COMPAS Experiments

The following implementation points are of note beyond those detailed in the paper:

- The data set was the same as used in [1] and was downloaded from `https://github.com/aaronjfisher/mcr-supplement`. The training and test sets as defined by [1] were used. The notebook file details the instructions to convert the data from that repository to csv for use in Python.

- Reference models were trained (included cross-validated meta-parameter search when applicable) using the training data. Reported model performance scores for the reference models were are for the test data set.

- In this work we consider the in-sample estimation of the MCR. While in-sample estimation is motivated in [1] the reported figures for grouped admissible and inadmissible variables in their evaluation based on the COMPAS data is based on sample-splitting. In order to verify the RBF-SVM implementation an exact replica of their experiment was run which confirmed the results they reported.

- The graphs generated by the Notebooks are per MCR estimation method, rather than the comparison graphs shown in the paper. This format was generated separately from the outputs. The information is the same.

- For RF MCR a Grid Search was performed to determine the optimal meta-parameters underpinned by 5-Fold cross-validation based on the training set. The reported MSE was computed on a held out test set. The grid search considered five different levels of minimum decrease in impurity and two different values for the number of features to be considered in each split (1 and the square root of the number of input features). Bootstrapping was disabled.

- For RBF-SVM the optimal meta-parameters found within the evaluation on this dataset by [1] were used. This was based on a meta-parameter search as per [1].

- The disentangling influence method utilized the same Random Forest reference model as RF-MCR. When learning the neural network as part of the variable importance procedure, the meta-parameters were set via the grid search method previously detailed. Diagnostic graphs for this process are shown in Figure 2. They indicate that the neural networks have done a reasonable, but imperfect, job of learning a disentangled representation. This provides some explanation as to the differing performance of this hybrid method compared to RF MCR+ and RBF-SVM MCR+. The error profile is not too dissimilar to those reported in the original authors work [4] in their empirical evaluations (on different datasets), and it is unclear if it is possible to learn a better representation. As discussed in the main paper, the requirement and time needed to tune this approach additionally makes its re-purposing to the use as an MCR+ measure less desirable (grid search took a several hours despite utilization of 10 cores). Further, as discussed in the section RF-MCR Analysis in the paper and below, the requirement to learn neural networks (even with fixed meta-parameters) as part of generating importance values results in run-times significantly greater than RF-MCR and RBF-SVM MCR. For COMPAS this took 20 minutes. In contrast RF-MCR runs in < 5 seconds. Note that in both the disentanglement method and RF MCR used the same trained reference model in the experiments.

Figure 2: Graph of diagnostic measures for the neural networks trained as part of the Disentangling Influence (INFL) method from [4] for the COMPAS experiments.

## 2.4 Breast Cancer Experiments

The following implementation points are of note beyond those detailed in the paper:

- The data set used was accessed directly from sklearn, via a corresponding API call (sklearn version 0.23.1). In order to reduce the number of variables, and allow clearer interpretation of results, a lightweight variable selection process was applied. This took the form of a traditional approach based on unconditional variable importance for a Random Forest re-run many times with different random seeds. This process took place *before* any methods reported in this work were considered. Input variables retained were: mean compactness, mean symmetry, area error, compactness error, symmetry error, worst radius, worst texture, worst perimeter, worst compactness, worst concave points and worst symmetry. The data was then randomly split into a training (66%) and test set (33%).

- Reference models were trained (included cross-validated meta-parameter search when applicable) using the training data. Reported model performance scores for the reference models were using the test data set, and achieved an accuracy of 95.74%.

- The reference model used in both RF-MCR and for the disentanglement influence method as part of the Hybrid-VI MCR was the same. An identical grid search for the forest meta-parameters was conducted as done for the COMPAS experiment. For conditional permutation importance (MCR- as part of the Hybrid-VI MCR) a new Random Forest is required to be trained in the R environment.

- As with the COMPAS experiments we report the in-sample estimation of the MCR.

- The disentangling influence method from [4], which is re-purposed to act as the MCR+ for the Hybrid-VI method, learnt the neural network's meta-parameters via the previously described grid search considering 80 different meta-parameterizations. The diagnostic graphs are shown in Figure 3. The graphs again indicate that although the networks have learnt relatively well, there remains per-feature error differences. Since these independently affect each variable's importance, it casts some doubts over the relative variable importances implied by this method, as discussed in the paper.

- The graphs generated in the Notebooks are per-method (rather than the comparison graphs collated and illustrated in the paper). This format was generated separately from the outputs. The information is the same.

Figure 3: Graph of diagnostic measures for the neural networks trained as part of the Disentangling Influence (INFL) method from [4] for the Cancer experiments. Variables are listed along the x-axis.

## 2.5 RF-MCR Analysis

Runtime analysis of RBF-SVM MCR and the proposed RF MCR with respect to the number of data points was conducted on the aforementioned machine with an Intel® Core™ i7-3930K CPU @ 3.20GHz with 64GB RAM and running Ubuntu 20.04, Python 3.6.10 and R 3.6.3. Both methods used single core algorithms. A comparison to the Hybrid-VI method was not made due to the method requiring the training of neural networks (and additionally the time to learn optimal meta-parameters for these networks) which meant the approach ran orders of magnitude slower (20mins vs. <5 seconds for a preliminary run on COMPAS). The runtime performance considered only the computation of the MCR (both MCR- and MCR+) and excluded the time taken to train the reference model in both cases, however, the reference model was retrained at each iteration. This is because the runtime of

the MCR algorithms are somewhat dependent on the complexity of the reference model structure. Simply increasing the dataset on which the MCR is computed would not be an accurate reflection of runtime in practice if a model trained on a smaller dataset was considered during a timing study. As such we consider the computation of in-sample MCR where the training dataset and the data set used to compute the MCR is of the same size at each iteration. A final note is that the two method differ slightly in the permutation scheme used. RBF-SVM MCR is coded in the implementation by [1] to use e-divide (Equation 3.4, pg. 9). In contrast in RF MCR we use repeated random permutations, repeating 10 times. The latter is more computationally expensive meaning the proposed method, RF MCR, is slightly disadvantaged. Given the results and the common place usage of repeated random permutations e-switch was not implemented and compared to within RF MCR.

The convergence of MCR+/- to a fixed point was additionally investigated on both the COMPAS and Breast Cancer data sets. For the COMPAS dataset an $\epsilon$ value of 3.1713 was used. To show convergence under different $\epsilon$, $\epsilon = 0.00001$ was used for the Breast Cancer dataset. In the paper, due to space constraints only two of the four graphs were presented, the convergence of MCR- for COMPAS and MCR+ for Cancer. The full four graphs are shown in Figures 4 - 7. Note, due to a different random seed being used during tree construction, the graphs are differ slightly to those in the full paper. The graphs show the same information, however, that quick convergence to a point of stability is achieved providing further evidence this occurs generally. Given the slight difference (exaggerated due to the change in scale) more trees may be justifiable for the COMPAS dataset, which might then enable the approach to report a slightly lower MCR- for age.

Figure 4: COMPAS dataset MCR- Results

Figure 5: COMPAS dataset MCR+ Results

Figure 6: Breast Cancer dataset MCR- results

Figure 7: Breast Cancer dataset MCR+ results

Finally an ablation study was conducted, in order to investigate the extent to which the two steps of the approach (surrogate replacement and the use of prediction equivalent trees, the $T^+$ transform) contribute to final estimation of MCR bounds. Specifically, we examined how much each step either increases (MCR+) or decreases (MCR-) the importance (from the reference model value) per variable on both the COMPAS and Cancer datasets. Note that, for reasons as listed in Section 4.3 in the paper, in the current implementation the epsilon parameter for surrogates is not implemented. The effect of the surrogate transform, therefore, is likely to under-report, particularly for larger $\epsilon$ values. To limit this effect $\epsilon$ was set to 0.1 for the ablation study. Results of the ablation study are shown in Figures

8 and 9, with both components evidenced as playing a notable role in determining the MCR- and MCR+ values.

Figure 8: Ablation Study results for the COMPAS dataset

Figure 9: Ablation Study results for the Breast Cancer dataset. For variables where the reference model's importance equalled the MCR- value no increase occurred and so no percentage change is shown.

# 3    Algorithm Reliance and Model Class Reliance

Algorithm reliance measures the predictive performance lost when training a model without one or more variables, compared to the predictive performance of a model trained with all variables included. By definition this is not a measure of MCR (see [1, §3.2]). [1, §9.1] consider two types of AR, *necessity* and *sufficiency*. Necessity is computed by training a model with all features (M0) and a second with all features except the feature of interest (M1). The difference in predictive performance is then measured. Sufficiency is computed by again considering M0, but also considering a model trained just on the variable of interest (M2). As [1] show empirically, AR necessity lower bounds MCR-. AR sufficiency, however, does not provide a good upper bound for MCR+, failing to capture and acknowledge the variables importance within interactions with other variables. Computing AR necessity and sufficiency for both the COMPAS and Cancer datasets show results inline with those reported by [1] when comparing AR and MCR. For COMPAS, AR necessity again lower bounds reported MCR- (trivially for charge degree, sex and race while for Age and priors count ratios of 1.66 and 1.074 are reported). For the Cancer dataset, since MCR- already identifies all variables as within some model not being required, the MCR- and AR are equivalent. AR sufficiency was also computed and was always less than the MCR+ bound as expected, highlighting its inability to attribute interaction effects to the variable.

# 4 Additional Empirical Analysis: Predicting negative birth outcomes

As part of the empirical analysis, our proposed methodology was also applied to a third dataset, with a focus on the application of method to a pressing real-world problem, rather than comparison to baselines. The data set selected is from an going project in East Africa looking at predicting negative birth outcomes based on surveyed data from: (1) community health worker visits taken before a birth, (2) living conditions and (3) features aggregated to a region level from alternative data sources such as mobile money records and call detail records. The output feature was a binary encoded variable indicating if the birth was subsequently a negative one or not.

The goal was to understand: (1) the predictive performance that could be obtained from modelling this issue (2) the potential utility of using (easier to collect) area-level and living condition features, under the assumption that the three feature types would likely share predictive information regarding the output variable. This information was then used as a basis for a real-world review into the type of information collected by the community health workers and as as basis for further future feature engineering in order to better develop the predictive model. Figure 10 shows the results of the RF-MCR analysis based on a reference model that achieved 66.5% accuracy on a held out test set.

In contrast to what would be seen by a traditional (e.g. permutation) variable importance analysis we see that region level features (home_ct, avg_trans_sent, avg_contacts, avg_sent_overall_trans, avg_trans_received) can range in importance, indicating well performing models exist where these features are utilized. This was equally true of the living condition features. Equally it showed that a number of survey questions did not seem to aid the prediction, with their almost zero MCR+ scores indicating they could (at least for this task) be considered for removal from the survey. The remaining survey questions (age, driver arranged in advance, partner permission, time of year of first visit, abortions) however show relatively large MCR- scores. These should therefore not be considered for exclusion from future surveying, as our MCR analysis indicates that they are indeed required in all models. An exception is perhaps the requirement to collect information on abortions (previously viewed as crucial), which might be revisited given the sensitive nature of the topic and the limited predictive impact indicated. Finally, results indicate that the derived features and living condition features share information about the output.

Figure 10: Negative Birth Outcomes dataset MCR Results

# 5  Proof

This section provides proof of the claim that:

$$MR_{X_1}(F_T^+, \Phi) \geq MR_{X_1}(F_S^+, \Phi) \tag{1}$$

Here $MR(.)$ is the model reliance of some random forest, $F$, on variable, $X_1$ given the context of $Z = \langle X_1, X_2, Y \rangle$. $X_2$ reflects the set of independent variables used in addition to $X_1$ by the forest to predict target variable, $Y$. Calculating MR(.) requires specification of a disruption function, $\Phi$, that is applied to $Z$, to generate a version of the variable interest, $X_1^\Phi \in Z^\Phi$, that is rendered as uninformative as possible in relation to $Y$. Finally, we recall that model reliance is defined as:

$$MR_{X_1}(F, \Phi) = L(F, Z^\Phi) - L(F, Z) \tag{2}$$
$$= L(F, \langle X_1^\Phi, X_2, Y \rangle) - L(F, \langle X_1, X_2, Y \rangle) \tag{3}$$

Below we show that following transformation of model $F_S^+$ to model $F_T^+$ (as per equation [9] in the full paper) model reliance on variable $X_1$ must be greater than or equal to its previous level.

We note for the proof that the transformation to $F_T^+$ is constructed so as to ensure *predictive equivalence* has be maintained. Specifically:

> *[The construction of $F_T^+$] is achieved by a second transformation $\mathcal{T}^+(F_S^+)$, mapping each tree in $F_S^+$ to some other in $F_S^+$ that produces the same predictions but which leverages our variable of interest the most (n.b. if a tree is already the most reliant on $X_1$ it is mapped to itself).*

Formally this means that, by construction:

$$\forall x \in Z, f \in F_S^+ : \; f(x) = \mathcal{T}(f(x)) \tag{4}$$
$$\implies \forall x \in Z : \; F_S^+(x) = F_T^+(x) \tag{5}$$
$$\implies L(F_S^+, Z) = L(F_T^+, Z) \tag{6}$$

Given these preliminaries, we first we substitute equation 3 into equation 1 to yield:

$$L(F_T^+, Z^\Phi) - L(F_T^+, Z) \geq L(F_S^+, Z^\Phi) - L(F_S^+, Z) \tag{7}$$
$$\implies L(F_T^+, Z^\Phi) \geq L(F_S^+, Z^\Phi) \tag{8}$$

Assuming that each $F$ reflects a random forest regressor, and that we are using a squared loss function to evaluate model reliance, then:

$$\implies \sum_{x^\Phi, y \in Z^\Phi} \left( (\frac{1}{|F_T^+|} \sum_{f_T \in F_T^+} f_T(x^\Phi)) - y \right)^2 \geq \sum_{x^\Phi, y \in Z^\Phi} \left( (\frac{1}{|F_S^+|} \sum_{f_S \in F_S^+} f_S(x^\Phi)) - y \right)^2 \tag{9}$$

Note that both the LHS and the RHS of this inequality now corresponds the total error of a Random Forest predictor against dataset $Z^\Phi$. This observation, however, allows us to leverage the *ambiguity decomposition* derived in [2], which guarantees the generalization error of each ensemble to be lower than the average generalization error of its constituents. Formally the decomposition states that for any random forest, $F$,:

$$L(F, Z) = E(F, Z) - A(F, Z) \tag{10}$$

where:

$$E(F, Z) = \frac{1}{|F|} \sum_{f \in F} L(f, Z) \tag{11}$$

$$A(F, Z) = \mathbb{E}_{\mathbb{Z}} \left[ \frac{1}{|F|} \sum_{f \in F} (f(x) - F^+(x))^2 \right] \tag{12}$$

$E(.)$ corresponds to the average generalization error of the individual trees in the forest, while the second term is the *ensemble ambiguity*, reflecting the variance of individual predictions made by the forest as a whole. Since A(.) is non-negative, the generalization error of the ensemble is therefore always smaller than equal to the average generalization error of its constituents [3][pg. 63].

We applying the ambiguity decomposition directly to both sides of equation 9:

$$\left[ \frac{1}{|Z|} \sum_{x^\Phi, y \in Z^\Phi} \frac{1}{|F_T^+|} \sum_{f_T} (f_T(x^\Phi) - y)^2 \right] - \left[ \frac{1}{|Z|} \sum_{x^\Phi, y \in Z^\Phi} \frac{1}{|F_T^+|} \sum_{f_T \in F_T} \left( f_T(x^\Phi) - F^+(x^\Phi) \right)^2 \right]$$
$$\geq$$
$$\left[ \frac{1}{|Z|} \sum_{x^\Phi, y \in Z^\Phi} \frac{1}{|F_S^+|} \sum_{f_S} (f_S(x^\Phi) - y)^2 \right] - \left[ \frac{1}{|Z|} \sum_{x^\Phi, y \in Z^\Phi} \frac{1}{|F_S^+|} \sum_{f_S \in F_S} \left( f_S(x^\Phi) - S^+(x^\Phi) \right)^2 \right]$$
$$(13)$$

Recognizing that the cardinality of the model doesn't change following transformation, such that $|F_T^+| = |F_S^+|$, and rearranging gives us:

$$\left[ \sum_{x^\Phi, y \in Z^\Phi} \sum_{f_T \in F_T^+} (f_T(x^\Phi) - y)^2 \right] - \left[ \sum_{x^\Phi, y \in Z^\Phi} \sum_{f_S \in F_S^+} (f_S(x^\Phi) - y)^2 \right] \geq$$
$$\left[ \sum_{x^\Phi, y \in Z^\Phi} \sum_{f_T \in F_T^+} \left( f_T(x^\Phi) - F_T^+(x^\Phi) \right)^2 \right] - \left[ \sum_{x^\Phi, y \in Z^\Phi} \sum_{f_S \in F_S} \left( f_S(x^\Phi) - F_S^+(x^\Phi) \right)^2 \right] \qquad (14)$$

This equation must always hold. This can be proven by considering the behaviour of each term on the left hand side of the equation (*LHS:T1*, *LHS:T2*) and right hand side (referred to as *RHS:T1*, *RHS:T2*), in light of potential bijective mappings of $F_T^+$ as it is transformed from $F_S^+$. Consider first the degenerative case where the transformation $\mathcal{T}^+ : F_S^+ \to F_S^+$, maps every element to itself so $F_T^+ = F_S^+$. In this case both terms on the LHS and RHS of equation 14 would be equivalent, both sides resolving to zero.

However, if the mapping $\mathcal{T}^+$ finds a tree that is substitutable for another in $F+_S$ (such a tree being predictively equivalent yet of greater model reliance), then while both *LHS:T2* and *RHS:T2* stay the same, both *LHS:T1* and *RHS:T1* must increase. This is guaranteed by construction, as the substitution undertaken by $\mathcal{T}^+$ only occurs if squared error against $Z^\Phi$) is increased (the definition of increased model reliance). Consequently equation 14 holds if and only if the resulting change in *LHS:T1* is greater than or equal to the *RHS:T1*:

$$\Delta \left[ \sum_{x^\Phi, y \in Z^\Phi} \sum_{f_T \in F_T^+} (f_T(x^\Phi) - y)^2 \right] \geq \Delta \left[ \sum_{x^\Phi, y \in Z^\Phi} \sum_{f_T \in F_T^+} \left( f_T(x^\Phi) - F_T^+(x^\Phi) \right)^2 \right] \qquad (15)$$

To prove this is always the case, consider that both components are of the form: $\sum_{f_S \in F_T^+} (f_T(x) - q)^2$. Minimizing this equation across all datapoints, given $q$ is a constant, gives us:

$$\frac{d}{dq} \left( \sum_{f \in F_T} (f(x) - q)^2 \right) = 0 \implies \tilde{q} = \frac{1}{|F|} f(x) = F_T^+(x)$$

Thus the right hand side term of equation 15 is minimized for any given, $Z^\Phi$ - and as a consequence the left hand side term can only be greater or equal to it. Therefore equation 15 holds, as does equation 14, and consequently also equation 8. Thus:

$$MR_{X_1}(F_T^+, \Phi) \geq MR_{X_1}(F_S^+, \Phi) \qquad (16)$$

This completes the proof.