[Reviews · NeurIPS 2020]

Review 1

Summary and Contributions: The paper presents a method for computing Model Class Reliance (a variable importance technique which measures variable importance for a set of models which are nearly as accurate as the best model) for the class of random forest classifiers and regressors. The superiority of the method vs. competing variable importance methods is shown on a synthetic dataset where the ground truth importances are known. Applications to 2 real world datasets are also presented. The algorithm runs in linearithmic time, which compares favorably with the complexity of MCR implementations for other model types.

Strengths: Variable importance is certainly an important topic which is gaining further attention as fairness and ethics issues receive additional focus in the current machine learning research environment. Random forests and related techniques such as XGBoost seem to be the strongest algorithms for the kind of structured, relatively low-dimensional tabular datasets which are typical in applications where fairness and ethics are major concerns, such as recidivism and loan approval (see for instance this Kaggle presentation for evidence of the success of random forests and related methods on this class of datasets: https://www.youtube.com/watch?v=jmHbS8z57yI). For those reasons, this paper seems to me to be very timely and relevant. The supplementary material is very thorough and it is clear that the authors have taken their reproducibility obligations very seriously.

Weaknesses: The main weakness I see have to do with the writing. I would suggest rewrites of many sentences (see below in the 'Clarity' section for specifics).

Correctness: I did not notice any incorrect claims. However, I did not have the time to go through the proofs in extreme detail, so I only have moderate confidence in my correctness assessments. It is certainly possible that I missed some errors in the proofs.

Clarity: There is a lot of copy editing to be done. Here are various suggested edits: Line 2 and 3: variables -> variable’s Line 27: comma needed: typically share predictive information regarding the output variable, [comma here] the learnt model potentially…. Line 37: under-pining -> underpinning Line 54: I suggest either hyphens or parentheses around ‘if exploratory analysis is being conducted’ Line 55: comma after ‘For instance’ Line 76 "this is an important practical issue"...should this be its own sentence? As it is written, this sentence does not make sense grammatically. Line 82 importance of a variable Line 88 being a measures -> being a measure Line 95 requirement renders or requirements render, not requirement render Line 134 con structed -> constructed Line 137 reflects on -> reflects Line 141 a all -> all Line 146 could be contributed our variable of interest -> could be contributed to our variable of interest Line 148 similarly manner -> in a similar manner Line 164 impossible iterate -> impossible to iterate Line 180 Where-> where…I would probably also get rid of the parentheses Line 239 is a the set -> is the set Line 236 ‘C is generated from B and perfectly colinear’ -> ‘C is generated so as to be perfectly colinear with B’ Line 348 in a way that significantly affect -> in a way that significantly affects Line 352 can form hypothesis -> can form hypotheses (or can form a hypothesis) Line 121 supplementary "however instead of leaning" -> learning

Relation to Prior Work: Relation to prior work is appropriately discussed.

Reproducibility: Yes

Additional Feedback: The other remark I have is more of a general remark about variable importance methods and ethics implications. It is certainly valuable to be able to (for instance) remove race as an input variable from a model without significantly degrading accuracy. However, even if you remove race, isn't it possible that other, remaining input variables implicitly predict race very accurately? If so, the model with race removed might still have an improper disparate impact on people of different races and might therefore effectively be discriminatory. Perhaps this concern is beyond the scope of this particular paper, but I am curious about whether the author(s) have any thoughts on this topic and whether either it is addressed by this method or whether it could be addressed by other methods. *** Update after reading other reviews and author feed *** The author feedback seems to me to be an adequate response to the concerns raised regarding MCR+ and MCR- benchmarks (one-feature vs zero-feature model and leave-one-feature-out) and also regarding the in-sample vs. out-of-sample issues (with the distinction being worthy of further investigation but the in-sample/asymptotic results being the current publication standard within the subfield). So I will keep my score at a 7.


Review 2

Summary and Contributions: This paper proposes a new method for estimating model class reliance values for random forests. Model class reliance (MCS) values are bounds on how much or how little models is a given class could rely on a feature while still maintaining the same accuracy. Their new method works by greedily moving trees towards or away from using a specific feature by swapping out/in that feature for split nodes. Then doing another greedy attempt to depend on or avoid all the trees in the ensemble with varying weights.

Strengths: The authors have gone to significant effort to develop a performant algorithm for MCR to make it practical for RF models. The algorithm seems to be well thought out and I assume works as claimed.

Weaknesses: The primary problem I have with this paper is that it does not compare with a good baseline. MCR- can be estimated by re-training a model without X_1 and then observing the change in loss (leave one out model). MCR+ can be roughly estimated by training a model on only X_1 and comparing that to the loss when no features are present (univariate model vs constant model). These are two classic ways to estimate the range of possible dependence on a feature. The authors say they don't use retraining methods because they are too slow. But that seems like a weak argument for RF models. Figure 4 shows that it takes 60 seconds to run their new method on a 5 variable model over a dataset with 2k samples. You could retrain an RF model 5-10 times very quickly on that sized data. So in summary, the point of the paper is the improve runtime with an approximate algorithm. In order to demonstrate value they need to show that this is better than the classic retraining method of measuring variable importance ranges. It either needs to be faster or more accurate (for some definition).

Correctness: I didn't notice any errors.

Clarity: Yes.

Relation to Prior Work: In a way yes, but see also my comments above about comparison with retraining methods. This is the big problem with the paper as it stands now.

Reproducibility: Yes

Additional Feedback: ===== Post response notes ===== Thank you for your careful and detailed reply. I can see why you took issue with the framing of the paper as a faster approximate algorithm. There is indeed a difference between making a tractable algorithm faster, and bringing something into the range of reasonable computation. The core issue I was concerned with however was not framing, but rather a lack of comparison with retraining-based methods for estimating the possible ranges of variable importance. My point was not that these are perfect estimators of MCR, but rather that they are tractable and the closest possible alternative, hence they should be compared with. See below for some notes the particular retraining methods I suggested: 1. MCR- with leave one out training. I agree that leave one out training will consider models that are outside the model performance band. But that is easily fixed by training a second model on the residual of the first (boosting), then you can scale the impact of the second model to control how much you want to depend on the variable vs. how much performance you are willing to lose. 2. MCR+ You are right that just training a univariate model could really miss a lot, but it does not need to be perfect to be a baseline. Just like above, if you need to stay within a performance band then train a second model with all features on the residuals and scale it to maximize dependence on the feature. Or slowly add other features with many residual models to get even more weight on the feature of interest. The point is not that these are perfect replacements, but they are practical for the datasets you use, and reflect the type of things people do in practice right now to measure the types of ranges of dependence given by MCR. I really liked your great algorithmic work! I just feel uncomfortable not having what I see as clear baseline not included in the results.


Review 3

Summary and Contributions: The authors present an algorithm for calculating model class reliance bounds for random forests. The algorithm builds on existing features of many random forest implementations, such as permutation importance and surrogate splits, and hence is fast and scalable. For a given random forest and feature set of interest X_1, to estimate MCR+, the algorithm modifies the trees such that they have maximum reliance on X_1, while preserving their predictions. This proceeds in two steps: first, whenever X_1 can be a surrogate variable in a split, it is promoted to being the splitting variable. Second, each tree in the forest is potentially replaced entirely by another tree that yields the same predictions, but leverages the X_1 more. Estimating MCR- is the opposite. The authors demonstrate the effectiveness of their method on a simulation of an XOR function, on the COMPAS model of recidivism, and on a Breast Cancer dataset from the UCI ML repository. They compare to other pointwise variable importance measures. For the COMPAS dataset, they compare to the RBF-SVM which calculates the [MCR-, MCR+] range using kernel methods. The RF MCR measures largely agree with the RBF-SVM MCR measures on the COMPAS dataset.

Strengths: The overall framework of estimating a [MCR-, MCR+] range for a class of models, rather than estimating pointwise VI for a single model, is a very powerful one. Model development involves many choices by the data scientist, and often is sensitive to random inputs like bootstrapping or non-determinism in optimization. This framework allows for estimating VI measures that are more stable to these sources of variation. The use of MCR- on the COMPAS example is a powerful illustration of how this method can help data scientists make better choices on what variables to include in a model. Since random forests are a dominant model paradigm, it is important to develop MCR estimation methods for them. The proposed algorithm is simple to understand, and relies on quantities that are generally already available during random forest model training (like surrogate splits). Hence it seems there are few barriers to it being adopted.

Weaknesses: The main concern I have with the paper is in the argument that the estimator does in fact converge to MCR+ and MCR- for random forests. Section 4.1 provides an argument that, as the number of trees goes to infinity, each tree will be replaced with one from its Rashomon set that is maximally dependent on X1 (when doing the MCR+ procedure). In finite samples, and with a finite number of trees, there are reasons to doubt whether this method provides consistent estimation of MCR for the random forest as a whole. The favorable generalization properties of random forests are known to be derived from the diversity of trees in the ensemble: a well known result of Breiman is that the generalization error decreases as the correlation of the residuals from the trees decreases. Both steps of the MCR procedure increase correlation of the trees. While the predictions of a tree and its surrogate may be identical for a given dataset, replacing a tree with the surrogate seems that it may decrease the expected generalization error of the tree as a whole. Thus, we may not stay in the ϵ-Rashomon set of the tree. Trees that have identical predictions on a given dataset still may be structurally quite different, and hence could change the overall generalization of the random forest. These issues need to be addressed in a more thorough theoretical investigation of the algorithm, as well as in simulations. I would also suggest the authors demonstrate an ablation experiment on the algorithm, to help the reader better understand the effect of the two different steps. In real data, it would seem that most surrogates are not exact, except perhaps near the leaves of the trees. Hence, the first step of the algorithm would leave the main structural properties of the trees intact, and would change permutation importance measures by a relatively small amount. The authors should also present simulation results on a simulation of larger scale than the XOR dataset. The XOR simulation is a very valuable one to include in the paper, as it helps the reader in understanding the performance of this method and the weakness of pointwise VI measures. However, I suggest there be a simulation that bridges the gap between the small-scale XOR dataset and a real higher dimensional, noisy problem. The authors could consider a random function from a Gaussian process prior, applied over covariates with a known covariance structure. Alternatively, the authors could simulate from a known causal DAG, and show that MCR- measures exclude certain non-causal variables.

Correctness: The technical details appear correct according to my knowledge.

Clarity: The presentation of the algorithm and the simulations are clear. As stated, I believe there should be more clarity in arguing that the algorithm indeed consistently estimates the MCR measures for the random forest as a whole.

Relation to Prior Work: The paper compares to a few dominant VI measures (INFL, SHAP). Since there are few works that target the MCR bounds explicitly, the comparisons given seem appropriate.

Reproducibility: Yes

Additional Feedback: *** Comments after reading author feedback *** Thank you to the authors for considering my comments and addressing them. I am glad to hear the authors are similarly interested in the differences between empirical and expected Rashomon sets. I would argue that this is particularly concerning for the RF use case, because the steps of re-correlating the trees undoes the variance-reducing property of bagging. The user might be left wondering, e.g. does the MCR- model really perform as well as it it would appear on future data. This could be addressed with another test set held out from original model training and MCR- calculations. That said, I think the paper still makes an interesting contribution. I would encourage the authors to indeed discuss this issue in the paper (perhaps if space is a concern, with extended discussion and/or experiments with an additional test set as suggested). Also, indeed the ablation experiments are interesting, and I would suggest the authors including them at least in the supplemental material if space is limited. In light of the feedback, I am raising my score to 7. ** End of comments (review text otherwise unchanged) *** The authors might discuss whether their algorithm can be applied to boosted tree methods. The first step, replacement of splits by surrogate variables, seems that it would immediately transfer. The second step, however, seems specific to random forest. Is the first step alone sufficient to get MCR bounds from boosted trees? A minor proofreading issue: in line 190, the transformation T^+ is typeset with a script T. Thereafter it is a math-mode T. These should be made consistent.


Review 4

Summary and Contributions: Rebuttal Response: I appreciate the response from the authors and their willingness to hone their narrative with proposed edits. Since I believe most of the edits are relatively minor (but important) clarifications, I am raising my score to a 7:accept. Model class reliance is a measure that helps determine variable importance. The authors introduce a computationally tractable method for estimating MCR of random forest models. Their work expands the class of models where it is possible to compute MCR/Rashomon sets. They demonstrate their methods on synthetic data, as well as two well-known, publicly available data sets.

Strengths: The authors develop an implementation strategy for a relatively new research thrust. I believe this work would be of interest to a broad portion of the NeurIPS community. The authors demonstrate the tractable nature of their proposed method. They show that the random forest class of model will weigh age highly during a classification task, which is a nice demonstration of their technique.

Weaknesses: The transformations in the discussion of MCR+ take on a large assumption. The authors are exploring a 'null space', for lack of a better term, where multiple models make identical predictions. While it's possible to find a structurally equivalent model on data we have seen before, it does not guarantee that the two models will perform identically on all future data. I think expanding on this would communicate an assumption that is needed for MCR. The comparison to SHAP is not entirely fair, in my opinion. SHAP is a model-centric explanation technique, where the goal is to explain what model M is doing. MCR is a population-centric explanation technique, where the goal is to explain how choice in algorithm (model class) affects the importance of the attributes across all possible models. To illustrate, let's say I want to understand what a deployed model learned...I would reach for SHAP before MCR. On the other hand, if I want to understand how my selection of model class influences what a model *could* learn, I would reach for MCR. I believe a clearer distinction between SHAP and MCR would also help better frame the authors' contributions. I would like to see more discussion on how to interpret the results. It is easy to see that 'Age' is a critical feature for random forest recidivism prediction, however, it's harder to extract meaning from the breast cancer data results. The discussion only points to 'variable importance applied to a single model will provide misleading results'. I disagree and believe that claim is misleading. I think there are times when it's important to look at variable importance for a single model and other times when the authors' MCR methods are the right choice. Perhaps consider swapping out the breast cancer data set for one that shows a nicer story, or consider going into more detail in the discussion.

Correctness: Decision trees, and resulting random forest models are piece-wise linear. This is an MCR technique for non-linear classifiers, inasmuch as piece-wise linear is not linear. Furthermore, the authors are computing MCR for an RBF kernel SVM, which is a non-linear model, even if it is only implemented for regression tasks. The statement on line 273 is vague and reads as an exaggeration. I would recommend clarifying. For instance, I think the description on lines 339-342 is more appropriate. The authors present the first MCR method for random forests, a non-linear model, in a classification + regression setting. Beyond this, the claims/methods in the paper appear correct/sound, to the best of my knowledge.

Clarity: Line 94: Pronouns confuse the intention of the sentence. I assume the first *they* refers to SHAP and the second *they* refers to MCR, but I would recommend making this clearer. In general, the paper reads well. Notation feels good, and the paper has a nice, linear story. I would like to see more discussion of the results, so the take-home message is a bit clearer.

Relation to Prior Work: The work is well motivated and has a clear trajectory from cited works. Perhaps a little more introduction to the concept of a Rashomon Set will make the paper more self-contained. My only suggestion here would be to make clearer the times when a variable importance technique like SHAP is more useful than MCR, and also when MCR is more useful than SHAP. This should help the broader NeurIPS audience clearly recognize when they should implement for the authors' work for their own systems/research.

Reproducibility: Yes

Additional Feedback: Line 146: could be attributed *to* our variable of interest Questions on interpreting results: Why are all the RF MCR (red) lines left-aligned at zero? It seems that it can't be possible for all attributes to exhibit 0.0 model reliance simultaneously. I understand this to mean that a random forest can perform well even after dropping any single attribute? What does it mean for RF MCR (red) 'worst perimeter' to have a range nearly double that of 'worst texture'? What is the significance of the difference of 'symmetry error' between red and gray lines? For broader impact, I largely agree. I would like to see some expansion on the last point. I think there is potential for using this technology adversarially. Perhaps consider a scenario where someone is trying to break a model. If they know which attributes are the most and least important to a particular model class, they could potentially craft adversarial samples that cause unexpected outcomes.

[Author Response · NeurIPS 2020]

**Reviewer 2:** We thank reviewer 2 for their positive comments and appreciate the helpful errata spotted. We agree the "Additional feedback" discussion is beyond the scope but look forward to having this important discussion.

**Reviewer 3: "So in summary, the point of the paper is the improve runtime with an approximate algorithm."** We strongly disagree with this casting, and the hope that the reviewer might reconsider in light of our comments. Improving runtime is not the key focus of the paper. There currently exists no algorithm to compute MCR for random forests whatsoever (and so nothing to improve runtime or accuracy over!). The approximations made in the technique we introduce are there so that calculation of MCR across a Rashomon set of RF models is physically tractable for the first time (along with accompanying proofs). This is the key contribution.

**"The primary problem I have with this paper is that it does not compare with a good baseline.":** We also disagree with this statement. The suggested MCR+ "rough equivalent" is inherently flawed and does not provide a sound measure of the upper bound of a variable when interactions exist amongst the input variables. E.g, in the synthetic XOR example each variable by itself predicts no better than chance, the same as a constant predictor. Therefore, the "rough equivalent MCR+" would return all zeros where the known truth is all 0.5 (see paper Fig 1). A method to measure upper bounds of variable importances in real world situations, which typically do contain variable interactions, is a key contribution of the work. More subtly the suggested MCR- baseline, hold-one-out VI, also does not offer a MCR baseline. As we note (although clearly opaquely) on lines 92-94 algorithm reliance methods (holding out variables) are not only a function of models that fit the data well while MCR methods are. The meaning and implication of this difference and why it matters (and therefore why hold-one-out should not be an MCR baseline) is strongly made in Fisher et al.'s (paper [7]) establishment of MCR in their 60-page JML paper both theoretically (§3.2) and empirically (§9.1). While a full discussion remains outside of the scope of this paper, given length constraints, we propose the inclusion of a slightly longer description of their argument to make it clearer why this should not be considered a baseline.

**Reviewer 4 and 5 question whether the found "equivalent" models will retain the same predictive performance on new data as the reference model (RM), i.e. remain in the $\epsilon-$Rashomon set around the RM as we tend to the population.** We agree this is an extremely interesting issue. However, the proposed method is based (inc. proofs and empirical evidence for MCR convergence) on constructing $\epsilon-$Rashomon sets ($\epsilon-$sets) based on in-sample (fit) equivalence. As correctly pointed out by the reviewers, making the claim that one can find $\epsilon-$sets based on generalized performance equivalence would require significantly more theoretical and empirical investigations. Its inclusion, in addition to developing and evaluating the proposed approach, however would take the work well beyond the length and scope possible in a NeurIPS paper. Therefore, we wish to note: (1) in general this is an open problem for MCR with Fisher et. al not fully addressing this issue but rather providing a proof (valid for our work) that in-sample estimation is sufficient under large sample sizes and a reference model with correctly selected complexity (their §4.1). (2) What is proposed corresponds to the real world use case where MCR would be computed based on a reference model trained on the full dataset as part of a fit(via CV to determine model complexity)-refit(on full data) methodology to model building. Notably, this is the underpinning use-case when the use of training data is motivated (instead of a test set) for computing traditional permutation importance (c.f. Interpretable Machine Learning by Molnar, 2020). (3) While we realise that our technique focuses on fit $\epsilon-$sets equivalence that we strongly believe this still makes an important statistical contribution in the road towards MCR. We thank the reviewers for highlighting this need for clarity and the importance of this discussion. We will include this clarification and discussion if accepted.

**Reviewer 4 suggests the inclusion of (1) an ablation experiment to understand the importance of the two steps (2) further simulation studies** With regard to (1) we have these results, as we did examined them ourselves, and simply left them out due to space. If accepted we will inject these as part of our additional page allowance. In brief: surrogates account for a change in the permutation importance by 0-13% while the majority (remainder) of the change comes from the second transform. With regards to (2), further simulation studies were run as part of the work and we agree that these provide additional insights. However, we feel that their inclusion would not provide significant additional insight worthy of the removal of other analysis/points given the available space. Finally, with respect to boosted trees - developing an MCR method for this class is of interest and part of our future work and something we're interested in discussing. Unfortunately, the approach does not directly transfer and is outside the scope of this work.

**Reviewer 5:** We agree the comparison to SHAP is not entirely fair. Line 90-92 attempts to indicate this. We are happy to adjust/extend the wording to clarify the use case differences between SHAP and MCR. With regard to the discussion of the cancer results: The fact that MCR- is 0 for all variables indicates that, there is at least one model in the Rashomon set (set of equally performant models) that does not rely on this variable to make predictions (although other models do, as indicated by the non-zero MCR+) and that the set is non-trivially large. Therefore, in certain contexts, i.e. when fitting a model and undertaking VIM to consider potential causal factors, fitting a single model would not provide the full picture. We agree, however, that in some cases (as mentioned in the introduction) this doesn't matter. If accepted we will briefly extend the discussion of the cancer results to ensure the interpretation is clear. We thank Reviewer 5 (as with all other reviewers) for the additional errata and pointing out lines requiring minor clarification (i.e. line 273) which we agree to.

[Meta-Review · NeurIPS 2020]

This is a relevant and timely paper that has been reviewed by four knowledgeable referees, who also thoroughly considered the author's response to their initial reviews. Three of these reviewers recommend acceptance, providing detailed suggestions on how to improve this work before its final submission. R3 recommends rejection. This dissenting opinion was upheld by R3 after discussion with other referees. R3 in my opinion correctly brings up that if the proposed approach aims to improve runtime with an approximate algorithm, this must be sufficiently demonstrated in experiments vs. straightforward alternatives (such as retraining-based methods). That has not been done in the original submission neither in the rebuttal. I find it necessary to include these empirical results, even though the straightforward alternatives feel inferior beforehand. I believe that R3 has a point here - it will be of value to the readers to see exactly how inferior to the proposed approach they might really be. I will recommend marginal acceptance, counting on the authors to follow recommendations and requests from the reviewers in the final copy of their paper.